# Genome Wide Identification of Sesame Dof Transcription Factors and Functional Analysis of SiDof8, SiDof10 and SiDof34 in Fatty Acid Synthesis

**DOI:** 10.3390/cimb47090700

**Published:** 2025-08-30

**Authors:** Feicui Zhang, Shanyu Chen, Feiling Song, Limin Shi, Xuegao Lv, Zhengmei Zhu, Huabing Lu

**Affiliations:** 1Institute of Maize and Featured Dryland Crops, Zhejiang Academy of Agricultural Sciences, Dongyang 322100, China; zhangfeicui77@126.com (F.Z.); dasoong@163.com (F.S.); shilimin821170@163.com (L.S.); lxg-909@163.com (X.L.); qiuxin-1980@163.com (Z.Z.); 2Institute of Crops and Nuclear Technology Utilization, Zhejiang Academy of Agricultural Sciences, Hangzhou 310004, China; csy6227862@163.com

**Keywords:** *Sesamum indicum*, *Dof* gene, fatty acid synthesis, whole genome identification

## Abstract

The Dof (DNA binding with one finger) protein is one of the unique transcription factors in plants, and it plays an important role in plant growth and stress response. Sesame is an oil-bearing crop with high oil content and rich nutrition. In this study, 34 *Dof* genes were identified in the sesame genome using bioinformatics technology, and their physicochemical properties, gene structure, conserved motifs, tissue-specific expression and functions in fatty acid synthesis were preliminarily analyzed. The results showed that although there were differences in sequence length, molecular weight and isoelectric point, SiDofs all contained a conservative zinc finger structure, which could be divided into three categories in phylogeny. All 34 *SiDof* genes contain 1–2 exons, and the conserved motifs among subfamilies are similar. Tissue-specific expression analysis showed that the expression levels of *SiDof8*, *SiDof10* and *SiDof34* were the highest in seeds 24 days after pollination. Overexpression of *SiDof8*, *SiDof10* and *SiDof34* could significantly increase the contents of C18:0, C18:1, C18:2 and C18:3, and all of them are located in the nucleus. There were Dof DNA binding elements in the promoter region of fatty acid synthesis genes. These results provide a theoretical basis for further study on the function of the sesame *Dof* genes and biological breeding.

## 1. Introduction

Dof belongs to a subfamily of the zinc finger protein family, which is a unique transcription factor in plants. The DNA binding domain of the Dof protein contains 52 amino acids, in which the motif of CX2CX21CX2C forms a single zinc finger structure, and four absolutely conserved Cys residues in this structure can covalently bind to Zn^2+^ to make it have the characteristics of binding to DNA [1]. All Dof proteins display similar DNA binding specificity, and its core DNA binding element is (A/T)AAAG or its reverse-oriented sequence, (T/A) CTTT [2]. Dof proteins are involved in stress response and biological processes such as plant tissue differentiation and seed development. For example, maize PBF (Dof), wheat WPBF (PBF-like) and barley BPBF (PBF-like) can regulate the expression of gliadin genes (endosperm expression genes) and thus affect seed development [3,4,5]; *Arabidopsis* AtDof3.7 and AtDof2.5 directly inhibit the biosynthesis of *GA3ox1*, thus reducing the content of gibberellin and inhibiting seed germination [6,7]; the combined actions of AtDof3.4 and AtDof2.3 can regulate the cyclin gene *CYCD3*, shorten the cell cycle, and reduce the number of cell divisions, resulting in plant dwarfism [8]; and MdDof54 can improve the drought tolerance of apple trees by increasing the photosynthesis rate and water transport capacity of branches during long-term drought [9].

Fatty acid metabolism is one of the most basic metabolic processes in plants, and its products can not only provide energy sources for the growth of plants but also provide polyunsaturated fatty acids for human beings which cannot be synthesized by themselves [10]. Moreover, a recent study suggests that fatty acids may also function as nitrification inhibitors or their precursors which can increase plant nitrogen-use efficiency [11]. The fatty acid synthases (FASs) in animals belong to type 1, which is a single functional protein with seven different catalytic regions [12], while in plants they belong to type 2, which is a complex of multiple catalytic enzymes [13], both using sucrose as the main carbon source. In plants, sucrose is oxidized to acetyl-CoA by the glycolytic pathway and then catalyzed by acetyl-CoA carboxylase (ACCase) to generate malonyl-CoA, which is combined with acyl carrier protein to generate malonyl-ACP [13]. The second step of this pathway is catalyzed by condensation enzymes 3-ketoacyl-ACP synthases (KASI and KASII) [14,15]. One molecule of acetyl-CoA and one molecule of malonyl-ACP undergo condensation, carbonyl reduction, dehydration and reduction again to produce fatty acids with 12–18 carbon atoms. Oleic acid is the branch point of fatty acid synthesis. On the one hand, linoleic acid and linolenic acid are produced by desaturase which is catalyzed by FAD2 and FAD3 [16,17] in endoplasmic reticulum or FAD6 [18] in chloroplast; the other further extends the production of longer chain fatty acids. Finally, the products synthesized from these fatty acids can be modified to produce triglycerides, which further compound into various lipids [19].

Some transcription factors have been reported to play important roles in oil synthesis. WRINKLED1(WRI, AP2/EREB family) can increase the oil content of seeds by 10–40% after overexpression in rape and maize [20,21]; soybean GmDof4 and GmDof11 promote oil synthesis by regulating the expression levels of *ACCase* and long-chain acyl-helper A synthetase genes [22]; overexpression of *GhDof1* can increase the oil content in cotton seeds [23]; and *CoDof30.1* was highly expressed in the middle stage of *Camellia oleifera* seed development and was involved in the fatty acid and lipid process [24].

Sesame is one of the oldest oil crops in the world. Sesame seeds are not only rich in oil and protein but also contain antioxidants, which have the effects of lowering cholesterol, regulating blood lipids, protecting the liver and kidney, protecting the cardiovascular system, and resisting inflammation and tumor growth, and they are widely used in food and health care products, medicine and other industries [25,26]. The content of unsaturated fatty acids in sesame seed oil is as high as 85%, which results in high nutritional and economic value. Therefore, it is of great significance to explore the regulation mechanism of sesame lipid synthesis. In this study, the whole genome of sesame Dof transcription factors was identified by bioinformatics methods, 34 members were screened, and their physicochemical properties, gene structure, conserved motifs and tissue expression were analyzed. It was found and proved for the first time that SiDof8, SiDof10 and SiDof34 could promote the contents of C18:0, C18:1, C18:2 and C18:3. These results lay a foundation for further understanding the biological functions of *Dof* genes in sesame and provide gene resources and theory for biological breeding of sesame.

## 2. Materials and Methods

### 2.1. Identification and Analysis of Physicochemical Properties of Dof Proteins

Based on the amino acid sequence (PF02701) of the Dof domain in the PFAM database (http://pfam.xfam.org/ (accessed on 16 April 2024)), the whole genome protein database of sesame was searched by Blastp (https://blast.ncbi.nlm.nih.gov/Blast.cgi, accessed on 16 April 2024), and all candidate sesame Dof protein sequences were compared. The screened sequences were submitted to Pfam for the search, and only C2-C2 domain sequences were retained; finally, the Dof family members were obtained. The physicochemical properties of SiDofs were analyzed by expasy (https://web.expasy.org/protparam/, accessed on 16 April 2024). Plant-mploc (http://www.csbio.sjtu.edu.cn/bioinf/plant-multi/ (accessed on 7 May 2024)) was used to predict the location of SiDofs.

### 2.2. Phylogenetic and Gene Structure Analysis

The Neighbor-joining algorithm (Bootstrap = 1000) of MEGA5.0 software was used to construct the phylogenetic tree, and other parameters were set by default values. The gtf format of the sesame genome annotation file was downloaded from NCBI, and TBtools (v2.320) was used to visualize the gene structure.

### 2.3. Conservative Motif Analysis

The online tool MEME (http://meme-suite.org/ (accessed on 9 May 2024)) was used to predict the conservative motifs of Dof proteins. The running parameters were default, “Select the site distribution” was set to “Any Number of Repetitions”, and the maximum number of motifs was set to 10. After that, TBtools was used for visualization.

### 2.4. Tissue Expression Analysis of Dof Genes

The roots, stems, leaves, and flowers, as well as seeds and carpels, of the local sesame species “Yiwu Black Sesame” at the initial flowering stage and 6 and 24 days after pollination were taken and ground in liquid nitrogen, and then the total RNA was extracted by using a plant total RNA extraction kit (TIANGEN, Beijing, China, DP432). cDNA was obtained by reverse transcription with the HiScript II Q RT SuperMix for Qpcr (Vazyme, Nanjing, China, R223) kit, and the reverse transcription product was diluted 5 times as a template. The primer designed by the “primer designing tool” of NCBI (*SiActin* accession number LOC105178124, which had the highest similarity with the *AtActin2* sequence) was used for quantitative fluorescence PCR analysis, with three biological replicates for each treatment, and the relative gene expression was analyzed with reference to the 2^−ΔΔCt^ method [27]. Primer information is listed in Appendix A.

### 2.5. Functional Analysis of SiDof8, SiDof10 and SiDof34 for Regulating Fatty Acid Synthesis

Primers SiDof8-OE, SiDof10-OE and SiDof34-OE (the primer sequences are listed in Appendix A) were used to amplify the full-length coding sequences of the three genes, and then they were connected to the pCAMBIA1300 vector and transformed into *Agrobacterium* GV3101. The positive monoclonal bacteria were shaken for 20 h. After centrifugation, the bacteria solution was resuspended to OD600 0.8 by using a proper amount of tobacco injection (0.01 M MES, 0.01 M MgCl_2_, 0.2 mM acetosyringone). The bacteria solution was placed in the dark at room temperature for 2 h and then injected into tobacco leaves (grown in a greenhouse at 24 °C for 6–7 weeks). Four days later, the injected tobacco leaves were cut off, and the content of fatty acid components was detected after quick freezing in liquid nitrogen.

We ground the sample into powder in a liquid nitrogen environment, added about 0.5 g of the sample into a 15 mL centrifuge tube with 5 mL of methanol and chloroform and soaked it for 2 h; then, we centrifuged the mixture at 10,000 rpm for 5 min before taking the supernatant and extracting once. We added 2.5 mL of the filtered supernatant into a 15 mL centrifuge tube, mixed well and blew it dry with a nitrogen blower. We added 1 mL of a 5% potassium hydroxide–methanol solution, shook it evenly, heated it at 60 °C for 30 min, and cooled it to room temperature. Then, we added 3 mL of a 14% boron trifluoride–methanol solution and repeated the above operation. Finally, we added 2.5 mL of n-hexane, and then after fully shaking, we took the supernatant for analysis by Agilent GC-MS (Santa Clara, California, USA) capillary DB-225ms, 30 m × 0.25 mm × 0.25 µm). The inlet was 280 °C and the split ratio was 20:1. The heating procedure was an initial temperature of 50 °C, 5 °C/min to 200 °C, and 2 °C/min to 230 °C, maintaining the temperature for 10 min. The conditions of mass spectrometry were 230 °C for ion source temperature, 150 °C for quadrupole temperature, ionization mode EI, 70 eV for electron energy, and scanning quality range from 35 *m*/*z* to 800 *m*/*z*. The injection volume was 1 microliter.

### 2.6. Subcellular Localization Analysis

The full-length coding sequence of *SiDof8*, *SiDof10* and *SiDof34* with the stop codon removed was amplified using primers SiDof8-GFP, SiDof10-GFP and SiDof34-GFP (the primer sequences are listed in Appendix A) and then connected to the pCAMBIA1300 vector and transformed into *Agrobacterium* GV3101. After shaking the bacteria, we used tobacco injection to resuspend the bacteria solution to OD600 0.8, and finally injected this into tobacco leaves. After 4 days of continuous growth in a greenhouse, we took photos with a laser confocal microscope.

## 3. Results

### 3.1. Analysis of Physicochemical Properties of Sesame Dof Proteins

Through bioinformatics analysis, 34 Dof members were identified in the sesame genome, which were named *SiDof1*-*SiDof34* in turn, and their characteristics such as chromosome start and end position, amino acid number, molecular weight, isoelectric point and subcellular localization were analyzed. From the distribution position of Scaffolds, there were seven *Dof* genes on LG8; four on LG3 and LG15; three on LG12; two on LG1 LG2, LG4, LG7, LG9 and LG11, respectively; and only one on LG5 and LG13. In addition, we did not determine the chromosome position of *SiDof30* and *SiDof31* (Appendix A). The amino acid number of sesame Dof proteins was between 155 aa (SiDof30) and 510 aa (SiDof27), and the molecular weight ranged from 17285.58 da (SiDof30) to 55383.27 da (SiDof27). The isoelectric point was between 4.479 (SiDof1) and 9.72 (SiDof25) (Appendix A). The results of subcellular localization prediction showed that all 34 SiDof members were located in the nucleus (Appendix A).

### 3.2. Genetic Structure and Phylogenetic Analysis of Sesame Dof Members

A phylogenetic tree of 34 SiDofs was constructed by the Neighbor-joining algorithm. As can be seen from Figure 1, 34 SiDofs could be divided into three groups, A, B and C, among which Group C was divided into two subgroups: C1 and C2. Group A had 7 members, Group B had only 2 members (SiDof23 and SiDof32), and Group C had the most with 25 members. At the same time, the gene structure was analyzed, and the results showed that all sesame *Dof* genes only contained one or two exons (Figure 1). Among them, in Group A, except for *SiDof15* and *SiDof30*, there was only one exon, and the other five members had two exons. In Group B, *SiDof23* had two exons, while *SiDof32* only contained one exon. In Group C, except for *SiDof16* and *SiDof33*, the other six members of the C1 subgroup contained only one exon. There were eight *Dof* genes in the C2 subgroup, including *SiDof2*, *SiDof31*, *SiDof8*, *SiDof12*, *SiDof4*, *SiDof20*, *SiDof21* and *SiDof34*, which contained two exons, and the other nine members only contained one exon (Figure 1).

### 3.3. Analysis of Conserved Motifs of Sesame Dof Members

The protein sequences of *Dof* genes were analyzed on the MEME website. The results showed that 10 conserved motifs (motif 1–motif 10) were found in the sequence of sesame Dof proteins, all of which contained motif 1, and the other conserved motifs were only present in some of the Dof proteins (Figure 2A,B). For example, motifs 2/3/4/5/6/7 were simultaneously distributed on SiDof3, SiDof7, SiDof14, SiDof19 and SiDof27, and 12 other Dof proteins, SiDof1, SiDof4, SiDof6, SiDof17, SiDof20, SiDof21, SiDof22, SiDof23, SiDof25, SiDof29, SiDof32 and SiDof34, only contained motif 1, without other conserved motifs (Figure 2A,B). Motif 1, also known as the C2-C2 conserved domain, was a unique DNA binding domain of Dof proteins. As can be seen from Figure 2C, the motif 1 sequence of 34 Dof proteins in sesame contains 23 conserved amino acids, in addition to four conserved cysteines.

### 3.4. Tissue-Specific Expression Analysis of Sesame Dof Genes

We used “Yiwu Black Sesame” as the material to explore the expression of *Dof* genes in different tissues of sesame. The results showed that the expression levels of *SiDof3*, *SiDof6*, *SiDof9*, *SiDof19* and *SiDof33* (Appendix A) were higher in sesame leaves. *SiDof8* was expressed in sesame roots and seeds 24 days after pollination (Figure 3A). *SiDof10* was only expressed in seeds 24 days after pollination (Figure 3B). The expression level of *SiDof12* in flowers was higher (Appendix A). It is worth noting that the expression of *SiDof34* (Figure 3C) in all tissues was significantly higher than other *SiDofs*. To sum up, the expression levels of *SiDof8*, *SiDof10* and *SiDof34* were highest in the seeds 24 days after pollination, suggesting that they might play a role in the development of sesame seeds.

### 3.5. SiDof8, SiDof10 and SiDof34 Were All Located in the Nucleus and Their DNA Binding Sites Were Included in the Promoter Region of Fatty Acid Synthesis-Related Genes

Subcellular localization analysis showed that SiDof8, SiDof10 and SiDof34 were localized in the nucleus (Appendix A). To clarify this result, we first used NLStradamus to predict whether its protein sequence contained nuclear localization signals. The results showed that SiDof8 and SiDof10 contained a nuclear localization signal, but SiDof34 had no signal (Figure 4A). Then the subcellular localization of SiDof8, SiDof10 and SiDof34 was analyzed in tobacco leaves. As can be seen from Figure 4B, compared with the empty vector expressed in the cell membrane, endoplasmic reticulum and nucleus, SiDof8, SiDof10 and SiDof34 fused with GFP were specifically expressed in the nucleus. In summary, the three proteins were probably playing their roles as transcription factors.

To further study the functions of *SiDof8*, *SiDof10* and *SiDof34*, we analyzed whether the promoter region of fatty acid synthesis-related genes contains Dof core DNA binding elements. The promoter sequences of *SiACCase* (LOC105165943), *SiFAD2* (LOC105163532), *SiSAD* (LOC105159421), *SiKA I* (LOC105175693) and *SiKA II* (LOC105173713) from NCBI (2000 bp upstream of the transcription start site) were downloaded and analyzed. The promoter regions of these five genes related to fatty acid synthesis all contained the core DNA binding elements A/TAAAG(CCCTT/A) of Dof genes, among which *SiACCase* and *SiKAS II* contained at most 18 and 12 core DNA binding elements, respectively, and *SiFAD2*, *SiSAD* and *SiKAS I* contained 5, 6 and 6 core binding elements, respectively (Figure 4C). This result shows that SiDof8, SiDof10 and SiDof34 have the possibility of directly binding to the promoter region of fatty acid synthesis-related genes to regulate fatty acid synthesis.

### 3.6. Overexpression of SiDof8, SiDof10 and SiDof34 Significantly Increased the Contents of C18:0, C18:1, C18:2 and C18:3

To verify the functions of *SiDof8*, *SiDof10* and *SiDof34* in fatty acid synthesis, we constructed their full-length coding sequences into the overexpression vector pCAMBIA1300 and injected them into tobacco leaves. Four days later, the tobacco leaves were analyzed. The results showed that the contents of C16:0 and C16:1 did not change significantly (Figure 5A,B). Meanwhile, the contents of C18:0, C18:1, C18:2 and C18:3 in overexpressing *SiDof10*, *SiDof8* and *SiDof34* tobacco leaves were significantly increased compared with the empty vector (Figure 5C–F). Among them, the contents of C18:0 increased by 20%, 39.3% and 29.5%, respectively (Figure 5C); the contents of C18:1 increased by 13.1%, 21.5% and 14.1%, respectively (Figure 5D); the contents of C18:2 increased by 29.4%, 33.9% and 52.8%, respectively (Figure 5E); and the contents of C18:3 increased by 31.5%, 79.4% and 44.3%, respectively (Figure 5F).

## 4. Discussion

The *Dof* gene family is one of the unique transcription factors in plants, and its members are characterized by a conserved C2-C2 finger motif in the DNA binding region. Since the first *Dof* gene was cloned in maize, its family members have been identified and analyzed in *Arabidopsis*, rape, sorghum and sunflower [28]. In this study, we identified 34 Dof proteins in sesame and analyzed their physicochemical properties, gene structure, protein structure and tissue expression. There were 37 members, 30 members, 117 members and 50 members in *Arabidopsis thaliana* [28], sorghum [29], rape [30] and sunflower [31], respectively. It can be seen that the number of *Dof* genes in sesame was slightly higher than that in sorghum. Sesame *Dof* genes contain 1–2 exons, and most protein structures only contain 1–3 conserved motifs (Figure 1 and Figure 2), which is consistent with the performance of *Dof* genes in sorghum, sunflower and rape.

The key genes involved in fatty acid synthesis and assembly have been reported in sesame. Wei et al. constructed genome haplotypes of 705 sesame varieties, combined with 56 agronomic traits in four environments, and analyzed the key genes in sesame oil synthesis, including *SiKASI*, *SiKASII*, *SiSAD*, *SiDGAT2* and *SiFATA* [32]. Hu Yaping cloned sesame *SiFAD2* and proved that it has the function of ω-6 desaturase [33]. In the sesame variety “Zhongzhi 13”, 52 *SiLTPs* (nonspecific lipid-transfer proteins) were identified and *SiLTPI.23* and *SiLTPI.28* were verified as candidate genes for the high oil content of sesame seeds; meanwhile, Dof transcription factors were found to interact with *SiLTPs* [34]. In this study, the promoter regions of five key genes (*SiACCase*, *SiFAD2*, *SiSAD*, *SiKAS I* and *SiKAS II*) for fatty acid synthesis in sesame were analyzed. The results showed that they all contained more than five core DNA binding elements of the Dof protein (Figure 4C), suggesting that SiDof8, SiDof10 and SiDof34 possibly promote fatty acid synthesis by directly regulating the expression of these genes. In other plants such as the non-edible energy crop Jatropha [35] and the high seed oil content crop Camelina [36], the promoter regions of key genes from FA and TAG biosynthetic pathways were all found in Dof DNA binding motifs. Therefore, Dof transcription factors may be widely involved in the process of lipid metabolism in a variety of oil crops.

The research results of Li et al. pointed out that the accumulation pattern of sesame fatty acids during seed development was S-shaped; that is, it slowly increases at first after pollination, then significantly increases at 24 days after pollination, then peaks at 36 days after pollination, and then declines [37]. *SiDof8*, *SiDof10* and *SiDof34* all had high expression levels in seeds 24 days after pollination (Figure 3), which was consistent with the time when the fatty acid content in seeds increased significantly, so it was speculated that the three genes might be involved in regulating fatty acid synthesis in sesame. Yeast, algae, protoplasts, tobacco and *Arabidopsis thaliana* are commonly used vectors to verify gene functions. Because sesame could not be genetically transformed, this study verified *SiDof8*, *SiDof10* and *SiDof34* functions by transferring these genes into tobacco leaves. Yin et al. showed that perilla PfSAD3 and PfSAD4 can complement the phenotype of yeast-related mutants, and overexpression of *PfSAD3* and *PfSAD4* in tobacco leaves increased the content of unsaturated fatty acids, indicating that tobacco leaves can be used to study whether the target gene is involved in fatty acid synthesis [38]. After overexpressing perilla *PfDof29* in tobacco, it was found that the contents of C16:0, C16:1 and C18:3 in tobacco leaves were significantly higher than those in the empty vector control, while the content of C18:2 was significantly lower [39]. In this study, the contents of C18:0, C18:1, C18:2 and C18:3 in overexpressed *SiDof8*, *SiDof10* and *SiDof34* tobacco leaves were significantly higher than those in the empty vector control, but the contents of C16:0 and C16:1 did not significantly change (Figure 5). From these results, it can be seen that SiDof8, SiDof10 and SiDof34 increased the total C18 content and also increased the unsaturated fatty acid content, which is partly in line with the function of *PfDof29*.

Sesame is not only high in oil content but also rich in nutrition. At present, the research on sesame oil synthesis and regulation focuses on synthesis-related genes, and the key genes regulating oil synthesis are poorly understood. In this study, three Dof transcription factors that can promote fatty acid synthesis were identified for the first time, and a large number of Dof DNA binding elements were found in the promoter region of the key genes of lipid synthesis. This result not only expands the theoretical basis of regulating sesame oil synthesis but also provides candidate genes for the subsequent improvement of sesame germplasm resources through gene editing.

## Figures and Tables

**Figure 1 cimb-47-00700-f001:**
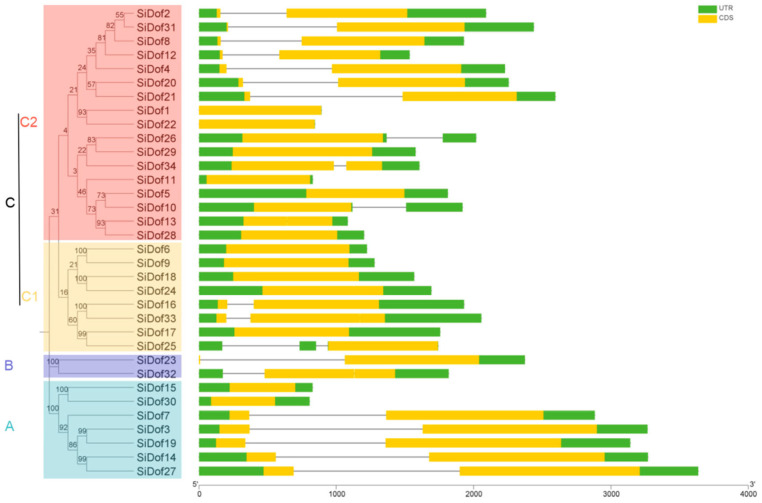
Phylogenetic and gene structure analysis of sesame Dof members.

**Figure 2 cimb-47-00700-f002:**
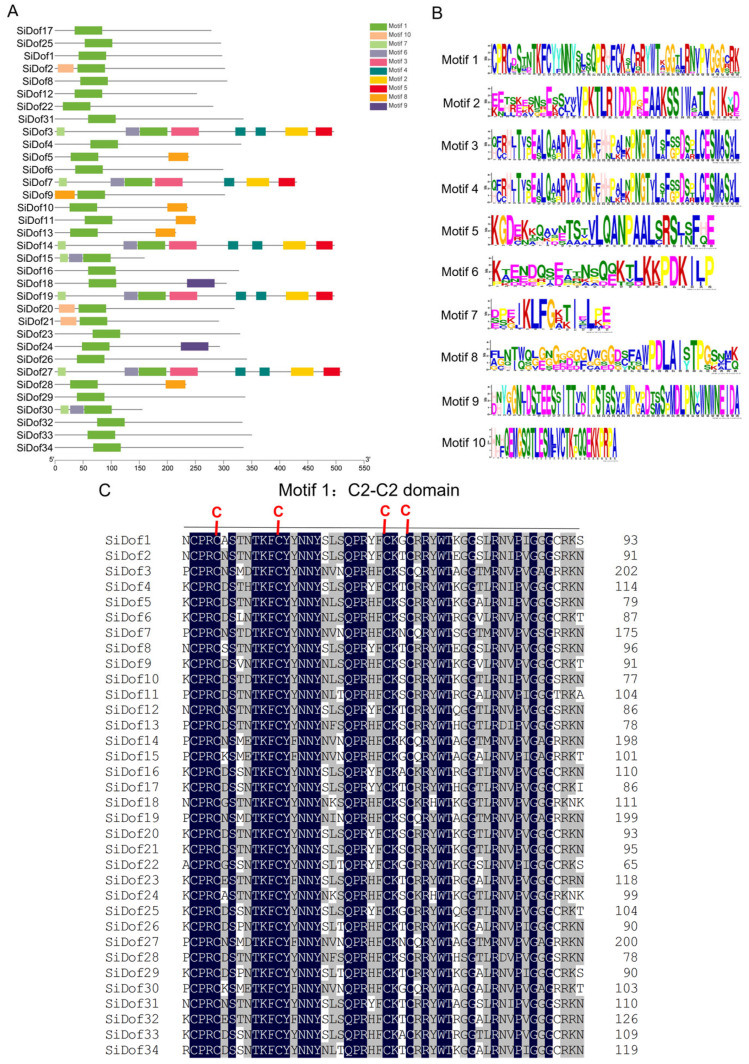
Sequence analysis of sesame Dof proteins. (**A**) Conserved motifs of 34 identified Dof proteins in sesame. (**B**) The sequence logo of motif 1–10. (**C**) Sequence alignment of conserved domains of 34 SiDofs. The red letter C indicate four conserved cysteines; Dark blue labeled letters indicate the same amino acid sites in all proteins; Grey labeled letters indicate amino acid sites with differences in 34 proteins.

**Figure 3 cimb-47-00700-f003:**
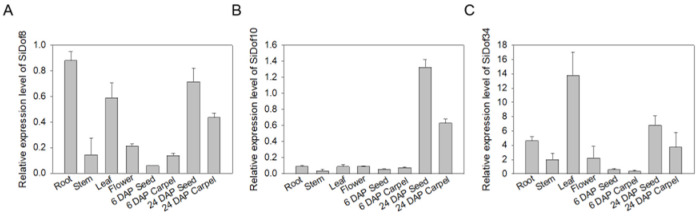
Relative expression analysis of *SiDof8* (**A**), *SiDof10* (**B**) and *SiDof34* (**C**) in different tissues of sesame.

**Figure 4 cimb-47-00700-f004:**
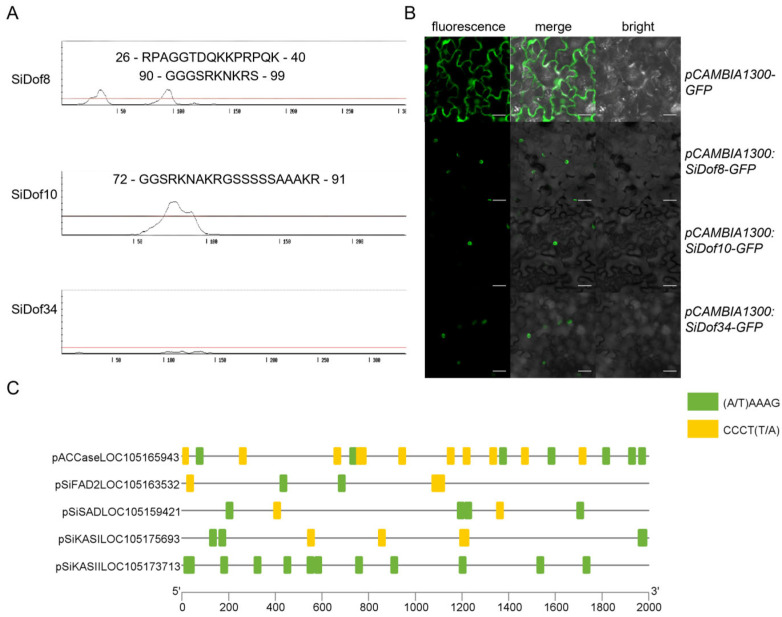
(**A**) Nuclear localization signal prediction of SiDof8, SiDof10 and SiDof34 in NLStradamus (http://www.moseslab.csb.utoronto.ca/NLStradamus/ (accessed on 5 July 2025)). (**B**) Subcellular localization analysis. Subcellular localization experiments were conducted with SiDof8 (second panel), SiDof10 (third panel) and SiDof34 (fourth panel) fused GFP as well as with GFP control vector (first panel). Bar = 0.02 cm (**C**) Analysis of Dof DNA binding elements in the promoter region of fatty acid synthesis genes. Yellow and green squares represented different Dof core DNA binding elements, respectively.

**Figure 5 cimb-47-00700-f005:**
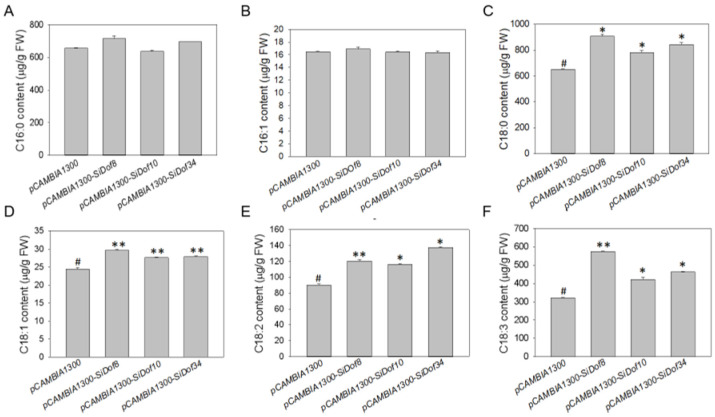
Analysis of fatty acid component content. (**A**–**F**) are C16:0, C16:1, C18:0, C18:1, C18:2 and C18:3, respectively. Error bar chart represents the standard deviation of three biological replicates; asterisks indicate significant differences relative to the control by two-tailed Student’s *t*-test. #, control; * *p* < 0.05, ** *p* < 0.001.

## Data Availability

Data is contained within the article and Appendix A.

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
