# Peer review of "Genome Wide Identification of Sesame Dof Transcription Factors and Functional Analysis of SiDof8, SiDof10 and SiDof34 in Fatty Acid Synthesis"

_cimb, 2025, doi:10.3390/cimb47090700_

Round 1

Reviewer 1 Report

Comments and Suggestions for Authors

The authors systematically analyzed the gene structure of the Dof transcription factor family in sesame (Sesamum indicum L.) and their expression abundance across different tissues. Furthermore, they conducted in-depth functional analyses of three members (SiDof8, SiDof10, and SiDof34) regarding their roles in fatty acid biosynthesis. However, the manuscript requires further revisions before publication.

Major Points:

  • In section 3.4, the authors mention "the expression of SiDof3, SiDof9 and SiDof19", but Figure 3B only displays results for SiDof19. I recommend including expression data for the other two genes.
  • Regarding section 3.4, while the heatmap in Figure 3A illustrates transcriptional patterns of 34 family members, I suggest incorporating expression data for SiDof22 and SiDof28 into the heatmap representation.
  • Section 3.5 presents a sudden transition to examining the effects of SiDof8, 10, and 34 overexpression on fatty acid content, creating a logical discontinuity. Subsequently, the authors analyze subcellular localization of these three proteins and examine promoter regions of key genes potentially involved in fatty acid biosynthesis. We recommend:

—Consolidating Figures 5 and 6 into a new Figure 4, with appropriate justification for selecting these three specific proteins

—Relegating current Figure 4 to become new Figure 5

—Reorganizing the narrative flow by either:

  1. a) Renaming panels A-B in current Figure 4 as E-F, or
  2. b) Moving the statement "In addition, the contents of C16:0 and C16:1 did not change significantly (Figure 4A and 4B)" to an earlier position

Additionally, the subcellular localization results would be more convincing with nuclear marker controls. I suggest including nuclear localization signal analysis of these protein sequences to support the findings.

  • The Introduction should be enhanced by:

—Providing more background on physiological functions of fatty acids and their metabolism in both plants and animals (e.g., Rice Science. 2024, 31(1): 87-102)

—Including additional literature support for linking Dof transcription factors with fatty acid metabolism

Minor Points:

  • The manuscript lacks line numbers, which complicates specific reference during review.
  • In Materials and Methods:

Section 2.4: "qrtsupermixforqpcr" requires proper formatting

Section 2.5: Please verify and correct subscript formatting

Author Response

Comments 1: In section 3.4, the authors mention "the expression of SiDof3, SiDof9 and SiDof19", but Figure 3B only displays results for SiDof19. I recommend including expression data for the other two genes.

Response 1: Thank you for pointing this out. I/We agree with this comment. The error has been corrected in the re-uploaded word document.

“The results showed that(Figure 3), the expression levels of SiDof19(Figure 3B), SiDof3, SiDof6, SiDof9, and SiDof33(Figure 3A) were higher in sesame leaves.”

Comments 2: Regarding section 3.4, while the heatmap in Figure 3A illustrates transcriptional patterns of 34 family members, I suggest incorporating expression data for SiDof22 and SiDof28 into the heatmap representation.

Response 2: Agree. The expression data for SiDof22 and SiDof28 was incorporated into the heatmap representation.

Comments 3: Section 3.5 presents a sudden transition to examining the effects of SiDof8, 10, and 34 overexpression on fatty acid content, creating a logical discontinuity. Subsequently, the authors analyze subcellular localization of these three proteins and examine promoter regions of key genes potentially involved in fatty acid biosynthesis. We recommend:

—Consolidating Figures 5 and 6 into a new Figure 4, with appropriate justification for selecting these three specific proteins

—Relegating current Figure 4 to become new Figure 5

—Reorganizing the narrative flow by either:

  1. a) Renaming panels A-B in current Figure 4 as E-F, or
  2. b) Moving the statement "In addition, the contents of C16:0 and C16:1 did not change significantly (Figure 4A and 4B)" to an earlier position

Response 3: Agree. Modified as recommended in the re-uploaded document.

3.5. SiDof8, SiDof10 and SiDof34 were all located in the nucleus and their DNA binding sites were included in the promoter region of fatty acid synthesis-related genes.

Subcellular localization analysis showed that SiDof8, SiDof10 and SiDof34 were localized in the nucleus(Table S3). To clarify this result, we used NLStradamus to predict whether its protein sequence contained nuclear localization signals firstly. The results showed that SiDof8 and SiDof10 contained nuclear localization signal, but SiDof34 had no signal(Figure S1). Secondly, The subcellular localization of SiDof8, SiDof10 and SiDof34 were analyzed in tobacco leaves. As can be seen from Figure 4A, compared with the empty vector expressed in cell membrane, endoplasmic reticulum and nucleus, the SiDof8, SiDof10 and SiDof34 fused with GFP were specifically expressed in the nucleus, indicating that the three proteins were indeed playing their roles as transcription factors.

To further study the functions of SiDof8, SiDof10 and SiDof34, we analyzed whether the promoter region of fatty acid synthesis-related genes contains Dof core DNA binding elements. The promoter sequences of SiACCase(LOC105165943), SiFAD2(LOC105163532), SiSAD(LOC105159421), SiKASâ… (LOC105175693) and SiKASâ…¡(LOC105173713) from NCBI(2000 bp upstream of the transcription start site) were dowloaded and analized.  The promoter regions of these five genes related to fatty acid synthesis all contained the core DNA binding elements A/TAAAG(CCCTT/A) of Dof genes, among which SiACCase and SiKASâ…¡ contained at most 18 and 12 core DNA binding elements respectively, and SiFAD2, SiSAD and SiKASâ…  contained 5, 6 and 6 core binding elements respectively(figure 4B). This result showed that SiDof8, SiDof10 and SiDof34 had the possibility of directly binding to the promoter region of fatty acid synthesis-related genes to regulate fatty acid synthesis.

Figure 4. (A)Subcellular localization analysis. Subcellular localization experiments were conducted with SiDof34(second panel), SiDof10(third panel) and SiDof8(fourth panel) fused GFP as well as with GFP control vector(first panel). (B)Analysis of Dofs DNA binding elements in the promoter region of fatty acid synthesis genes.Yellow and green squares represented different Dof core DNA binding elements, respectively.

3.6. Overexpression of SiDof8, SiDof10 and SiDof34 significantly increased the contents of C18:0, C18:1, C18:2 and C18:3.

So as to verify the functions of SiDof8, SiDof10 and SiDof34 in fatty acid synthesis, we constructed their full-length coding sequences into the overexpression vector pCAMBIA1300, and injected them into tobacco leaves. Four days later, the tobacco leaves were detected respectively. The results showed that the contents of C16:0 and C16:1 did not change significantly(Figure 5A and 5B). Meanwhile, the contents of C18:0, C18:1, C18:2 and C18:3 in overexpressing SiDof10, SiDof8 and SiDof34 tobacco leaves were significantly increased compared with the empty vector(Figure 5C-F). Among them, the contents of C18:0 increased by 20%, 39.3% and 29.5% respectively(Figure 5C), the contents of C18:1 increased by 13.1%, 21.5% and 14.1% respectively(Figure 5D), the contents of C18:2 increased by 29.4%, 33.9% and 52.8% respectively(Figure 5E), as well as the contents of C18:3 increased by 31.5%, 79.4% and 44.3% respectively(Figure 5F).  

Figure 5. Analysis of fatty acid component content, (A-F) are the C16:0, C16:1, C18:0, C18:1, C18:2 and C18:3, respectively. Error bar chart represents the standard deviation of three biological replicates; asterisks indicate significant differences relative to the control by a two-tailed Student’s t-test. #, control; * p < 0.05, ** p < 0.001.

Comments 4: Additionally, the subcellular localization results would be more convincing with nuclear marker controls. I suggest including nuclear localization signal analysis of these protein sequences to support the findings.

Response 4: Agree. I used NLStradamus to predict nuclear localization signal analysis of SiDof8, 10, and 34  protein sequences. The results showed that SiDof8 and SiDof10 contained nuclear localization signal, but SiDof34 had no signal(Figure S1).

Comments 5:The Introduction should be enhanced by:

—Providing more background on physiological functions of fatty acids and their metabolism in both plants and animals (e.g., Rice Science. 2024, 31(1): 87-102)

Response 5: Agree. 

“Fatty acid metabolism is one of the most basic metabolic processes in plants, and its products can not only provide energy sources for the germination and growth of plants, but also provide polyunsaturated fatty acids for human beings which cannot be synthesized by themselves[10]. The fatty acid synthases (FASs) in animals belong to type 1, which is a single functional protein with seven different catalytic regions[11], while in plants belong to type 2, which is a complex of multiple catalytic enzymes[12], both using sucrose as the main carbon source. In plants, Sucrose is oxidized to acetyl-CoA by glycolytic pathway, and then catalyzed by acetyl-CoA carboxylase(ACCase) to generate malonyl-CoA, which is combined with acyl carrier protein to generate malonyl-ACP[12]. The second step of this pathway is catalyzed by condensation enzymes 3- ketoacyl -ACP synthases(KASI and KASII)[13,14]. One molecule of acetyl-CoA and one molecule of malonyl-ACP undergo condensation, carbonyl reduction, dehydration and reduction again to produce fatty acids with 12-18 carbon atoms. Oleic acid is the branch point of fatty acid synthesis. On the one hand, linoleic acid and linolenic acid are produced by desaturase which are catalyzed by FAD2 and FAD3[15,16] in endoplasmic reticulum or FAD6[17] in chloroplast; the other is to further extend the production of longer chain fatty acids. Finally, the products synthesized from these fatty acids can be modified to produce triglycerides, which further compounding into various lipids[18]. ”

Comments 6:—Including additional literature support for linking Dof transcription factors with fatty acid metabolism

Response 6: Agree. Few transcription factors have been reported to participate in fatty acid metabolism, of which only three Dof transcription factors are clearly involved in fatty acid synthesis. I added one.

“23. Zhao, G,; Chen, L.; Zhang, L.; Liao, X.; Wang, J. Identification and expression profiling of the CoDof genes involved in fatty acid/lipid biosynthesis of tetraploid Camellia oleifera. Front Plant Sci. 2025, 16:1599849.”

Comments 7: Section 2.4: "qrtsupermixforqpcr" requires proper formatting

Response 7: Agree. The error has been corrected in the re-uploaded word document.

“HiScriptâ…¡Q RT SuperMix for qPCR”

Comments 8: Section 2.5: Please verify and correct subscript formatting

Response 8: Agree. The error has been corrected in the re-uploaded word document.

Reviewer 2 Report

Comments and Suggestions for Authors

The paper was concentrated on the genome wide identification of sesame Dof transcription factors and functional analysis of SiDof8, SiDof10 and SiDof34 in fatty acid synthesis. The results indicated that all 34 SiDof genes contain 1-2 exons, and the conserved motifs among subfamilies are similar. Tissue-specific expression analysis showed that the expression levels of SiDof8, SiDof10 and SiDof34 were the highest in seeds 24 days after pollination. Overexpression of SiDof8, SiDof10 and SiDof34 could significantly increase the contents of C18:0, C18:1, C18:2 and C18:3, and all of them are located in the nucleus. In addition, there were Dof DNA binding elements in the promoter region of fatty acid synthesis genes. The results are interesting, but the layout of the manuscript and few fragments of text need to be considerably improved.

I recommend the following minor revisions of the manuscript before further editorial processing:

- I suggest removing Table 1 to Supplementary File.

- The caption for Figure 3 requires completion by including the type of statistical test used and the reference gene.

- The purpose and novelty of the study should be described more precisely.

- The discussion section should be expanded and deepened, with a more thorough interpretation of the obtained results.

- The Authors should add Table S1 (mentioned in the text), including the primer sequences.

- The citations within the scope of the research topic should be expanded, including the incorporation of more recent and relevant literature to strengthen the scientific context and support the discussion.

Author Response

Comments 1: - I suggest removing Table 1 to Supplementary File.

Response 1: Thank you for pointing this out. I agree with this comment. Modified as recommended in the re-uploaded word document. Table 1 was modified as Table S1.

Comments 2: - The caption for Figure 3 requires completion by including the type of statistical test used and the reference gene.

Response 2: I think what you said should be the original figure 4 (now changed to figure 5) and I added sentence “Asterisks indicate significant differences relative to the control by a two-tailed Student’s t-test. #, control; * p < 0.05, ** p < 0.001.” to supplement.

Comments 3: - The purpose and novelty of the study should be described more precisely.

Response 3: Agree. Described in disscussion:

“Sesame is not only high in oil content but also rich in nutrition. At present, the research on sesame oil synthesis and regulation focuses on the synthesis related genes, and the key genes regulating oil synthesis are poorly understood. In this study, three Dof transcription factors that can promote fatty acid synthesis were identified for the first time, and a large number of Dof DNA binding elements were found in the promoter region of the key genes of lipid synthesis. This result not only expands the theoretical basis of regulating sesame oil synthesis, but also provides candidate genes for the subsequent improvement of sesame germplasm resources through gene editing.”

Comments 4: - The discussion section should be expanded and deepened, with a more thorough interpretation of the obtained results.

Response 4: Agree. Modified as recommended in the re-uploaded document.

Comments 5: - The Authors should add Table S1 (mentioned in the text), including the primer sequences.

Response 5: Agree. I added in the re-uploaded document.

Comments 6: - The citations within the scope of the research topic should be expanded, including the incorporation of more recent and relevant literature to strengthen the scientific context and support the discussion.

Response 6: Agree. Modified as recommended in the re-uploaded document.

Round 2

Reviewer 1 Report

Comments and Suggestions for Authors

Major points:

1) The Chinese characters appearing in Figure 2C should be removed.

2) In Figure 3, where the authors begin investigating the potential roles of SiDof8, SiDof10, and SiDof34 in regulating fatty acid synthesis, the inclusion of SiDof19 results in Figure 3B appears unnecessary. We recommend relocating these results to the supplementary materials and instead including tissue expression data for SiDof8 and SiDof10 in this section, which would provide better logical flow.

3) For Figure 4, the absence of nuclear localization markers renders the results in Figure 4A insufficiently supported. We suggest incorporating the nuclear localization signal analysis for these three proteins directly in Figure 4, rather than in supplementary materials, to provide stronger evidence for their nuclear localization.

4) In Figure 5, what is the rationale for presenting SiDof10, Dof8, and Dof34 in this particular order? Unless there is a specific reason, we recommend arranging them in sequential order for consistency.

Minor points:

1) In section 3.5, the statement "...indicating that the three proteins were indeed playing their roles as transcription factors" is not accurate. The current evidence only demonstrates nuclear localization of these proteins without transcriptional activity verification. The conclusion should be moderated to indicate that they may potentially function as transcription factors.

2) The work by Di et al. (Rice Science. 2024, 31(1): 87-102) provides experimental evidence suggesting that fatty acids may function as nitrification inhibitors or their precursors. This relevant literature should be cited and discussed in the Introduction section.

Author Response

Major points:

1) The Chinese characters appearing in Figure 2C should be removed.

Response 1: Thank you for pointing this out. The error has been corrected in the re-uploaded word document.

2) In Figure 3, where the authors begin investigating the potential roles of SiDof8, SiDof10, and SiDof34 in regulating fatty acid synthesis, the inclusion of SiDof19 results in Figure 3B appears unnecessary. We recommend relocating these results to the supplementary materials and instead including tissue expression data for SiDof8 and SiDof10 in this section, which would provide better logical flow.

Response 2: Agree. Modified as recommended in the re-uploaded document. I put the tissue expression data for SiDof8, SiDof10 and SiDof34 in Figure 3, and the tissue expression data for other SiDofs in Figure S1.

3) For Figure 4, the absence of nuclear localization markers renders the results in Figure 4A insufficiently supported. We suggest incorporating the nuclear localization signal analysis for these three proteins directly in Figure 4, rather than in supplementary materials, to provide stronger evidence for their nuclear localization.

Response 3: Agree. Modified as recommended in the re-uploaded document.

4) In Figure 5, what is the rationale for presenting SiDof10, Dof8, and Dof34 in this particular order? Unless there is a specific reason, we recommend arranging them in sequential order for consistency.

Response 4: Agree. Modified as recommended in the re-uploaded document.

Minor points:

1) In section 3.5, the statement "...indicating that the three proteins were indeed playing their roles as transcription factors" is not accurate. The current evidence only demonstrates nuclear localization of these proteins without transcriptional activity verification. The conclusion should be moderated to indicate that they may potentially function as transcription factors.

Response 1: Agree. Modified as recommended in the re-uploaded document.

“Then the subcellular localization of SiDof8, SiDof10 and SiDof34 were analyzed in tobacco leaves. As can be seen from Figure 4B, compared with the empty vector expressed in cell membrane, endoplasmic reticulum and nucleus, the SiDof8, SiDof10 and SiDof34 fused with GFP were specifically expressed in the nucleus. In a word, the three proteins were probably playing their roles as transcription factors. ”

2) The work by Di et al. (Rice Science. 2024, 31(1): 87-102) provides experimental evidence suggesting that fatty acids may function as nitrification inhibitors or their precursors. This relevant literature should be cited and discussed in the Introduction section.

Response 2: Agree. Modified as recommended in the re-uploaded document.

“Moreover, a recent study suggests that fatty acids may also function as nitrification inhibitors or their precursors which can increase plant nitrogen- use efficiency[11]. ”

Round 3

Reviewer 1 Report

Comments and Suggestions for Authors

The authors have answered all my concerns.